

# Stratospheric aerosol size reduction after volcanic eruptions

Felix Wrana[1], Ulrike Niemeier[2], Larry W. Thomason[3], Sandra Wallis[1], and Christian von Savigny[1]

[1]Institute of Physics, University of Greifswald, Felix-Hausdorff-Str. 6, 17489 Greifswald, Germany
[2]Max Planck Institute for Meteorology, Bundesstr. 53, 20146 Hamburg, Germany
[3]NASA Langley Research Center, Hampton, Virginia, USA

**Correspondence:** Felix Wrana
(felix.wrana@uni-greifswald.de)

**Abstract.**

The stratospheric aerosol layer plays an important role in the radiative balance of earth primarily through scattering of solar radiation. The magnitude of this effect depends critically on the size distribution of the aerosols. The aerosol layer is in large part fed by volcanic eruptions strong enough to inject gaseous sulfur species into the stratosphere. The evolution of the

stratospheric aerosol size after volcanic eruptions is currently one of the biggest uncertainties in stratospheric aerosol science. We retrieved aerosol particle size information from satellite solar occultation measurements from the Stratospheric Aerosol and Gas Experiment III mounted on the International Space Station (SAGE III/ISS) using a robust spectral method. We show that, surprisingly, some volcanic eruptions can lead to a decrease in average aerosol size, like the 2018 Ambae and the 2021 La Soufrière eruptions. In 2019 an intriguing contrast is observed, where the Raikoke eruption (48°N, 153°E) in 2019 led to

the more expected stratospheric aerosol size increase, while the Ulawun eruptions (5°S, 151°E), which followed shortly after, again resulted in a reduction of the median radius and absolute mode width values in the lowermost stratosphere. In addition, the Raikoke and Ulawun eruptions were simulated with the aerosol climate model MAECHAM5-HAM. In these model runs, the evolution of the extinction coefficient as well as of the effective radius could be reproduced well for the first 3 months of volcanic activity. However, the long lifetime of the very small aerosol sizes of many months observed in the satellite retrieval

data could not be reproduced.

## 1   Introduction

The variability of stratospheric sulfate aerosol is still not well understood and the question of whether they increase in size after large $SO_2$ injections, e.g. by volcanic eruptions reaching the stratosphere, is one of the most important research questions of

recent years (Robock, 2015). The size of stratospheric aerosol is a crucial factor for their effect on the lifetime of the aerosols and atmospheric chemistry (Deshler, 2008; Kremser et al., 2016), e.g. on ozone levels, as well as for their effect on the radiative balance of earth and therefore their net cooling effect on Earth's surface (Lacis et al., 1992). The expectation of how the size





distribution of stratospheric aerosol changes after volcanic injections of sulfurous gases into the stratosphere is still largely based on studies on the Mt. Pinatubo eruption in 1991, which led to a significant increase in the size of stratospheric sulfate

aerosols (Bingen et al., 2004; Deshler et al., 2003; Deshler, 2008). It was the largest volcanic eruption observed with satellite instruments to date and had a strong impact on the stratospheric aerosol distribution. Because of this, the observations of this eruption are widely used to evaluate aerosol microphysical models (Timmreck, 2001; Aquila et al., 2012; Niemeier et al., 2009; Sukhodolov et al., 2018; Quaglia et al., 2023).

On the other hand, many much smaller volcanic events have been observed by space-based instruments over the past 40

years. Those smaller eruptions may have different effects on the stratospheric aerosol size. In this work, as a part of the research project VolImpact (von Savigny et al., 2020a), we investigate the evolution of the stratospheric aerosol size after the eruptions of four volcanoes within the mission time of the Stratospheric Aerosol and Gas Experiment III mounted on the International Space Station (SAGE III/ISS), namely Ambae (15°S, 168°E), Raikoke (48°N, 153°E), Ulawun (5°S, 151°E) and La Soufrière (13°N, 61°W).

Using the remote sensing data set of the SAGE III instrument mounted on the International Space Station (SAGE III/ISS) we retrieved the size distribution parameters of monomodal lognormal size distributions with a robust multi wavelength method. We show that while the Raikoke eruption had an increasing effect on the average aerosol size, the Ambae, Ulawun and La Soufrière eruptions led to an unexpected and considerable decrease. We also simulated the Raikoke and Ulawun eruptions using the aerosol climate model MAECHAM5-HAM (short ECHAM) (Stier et al., 2005; Niemeier et al., 2009), in order

to investigate whether it can be used to identify and understand the main dynamical and microphysical factors controlling the aerosol size evolution caused by the eruptions. This is also relevant for the modelling community, since previous model studies, usually concentrated on the long term development of the particle size (Sukhodolov et al., 2018) instead of on the first few months of its evolution.

In Sect. 2 the instrumental and modelling data sets used and the methods employed are described. In Sect. 3 an overview

over the three periods of volcanic activity that are being investigated is given, followed by the presentation of the spatial and temporal evolution of the retrieved aerosol size distribution parameters for those periods (Sect. 4), which is discussed in Sect. 5. Finally, in Sect. 6 model simulations of aerosol extinction and size for the Raikoke and Ulawun eruptive period are presented and compared to the SAGE III/ISS retrieval data.

## 2    Instruments and Methodology

### 2.1    SAGE III/ISS instrument

The SAGE III/ISS instrument is the latest successor of the previous SAM II, SAGE I, SAGE II, and SAGE III Meteor-3M satellite experiments and aims at the investigation of the stratosphere and upper troposphere. Its mission started in June 2017 with its measurements still ongoing at the time of writing. Onboard the ISS, which has an orbital inclination of 51.6 ° and orbits Earth in about 92 minutes, the instrument is performing lunar and solar occultation measurements. Only the latter are used in

this work. Due to the platform's orbit SAGE III/ISS observes roughly 15 sunrise and 15 sunset events in 24 hours. Sunrise





and sunset measurements are taken at different latitudes and these latitudes oscillate roughly between $70\,°N$ and $70\,°S$ with a period of around 2 months (Cisewski et al., 2014).

Through the measurement of the solar radiation attenuated by atmospheric constituents the SAGE III/ISS data set provides information on gases like ozone and water vapour as well as on aerosols. Aerosol extinction coefficients are provided at 9 spectral channels between $384\,nm$ and $1544\,nm$ on a 0.5 km grid from the Earth's surface up to 45 km altitude. The spectral resolution of the first 8 channels between $384\,nm$ and $1020nm$, which are all covered by a CCD array is $1-2\,nm$. The $1544\,nm$ channel is detected by an indium gallium arsenide (InGaAs) infrared photodiode and has a band-width of about $30\,nm$. In this work, version 5.21 of the SAGE III/ISS level 2 data is used (NASA, 2023).

Advantages of using aerosol extinction data from the SAGE III/ISS satellite solar occultation measurements are firstly that there is no need to make assumptions on the particle size distribution to retrieve the extinction coefficients in the first place. Secondly, the sun provides a strong signal, benefitting the signal to noise ratio of the data product. Data is provided with a high vertical resolution. Additionally, since the measurements of every sunset or sunrise event are calibrated with corresponding direct solar irradiance measurements between $100\,km$ and $300\,km$, the data set is relatively unaffected by changes in the instrument over time. On the other hand the spatial and temporal coverage is low compared to, e.g. satellite limb measurements. This is because in one day only around 30 profiles are obtained and only a narrow latitude range is covered. Still, compared to ground based measurements, an almost global coverage is possible within one to two months. Therefore, an analysis of the spatial and temporal evolution of quantities derived from the SAGE III/ISS measurements is feasible (see Sect. 3 and 4) (SAGE III/ISS Users Guide, 2022).

## 2.2 Particle size retrieval method

The method that was used for the retrieval of the stratospheric aerosol particle size from the SAGE III/ISS data set has been described in detail in Wrana et al. (2021). Therefore, the size distribution parameter retrieval method employed will only be described briefly here.

A monomodal lognormal shape of the size distribution for stratospheric aerosols is assumed, which is expressed mathematically as follows:

$$\frac{dN(r)}{dr} = \frac{N_0}{\sqrt{2\pi} \cdot r \cdot ln\sigma} \cdot exp(-\frac{ln^2(r/r_{med})}{2ln^2\sigma}) \tag{1}$$

with $N_0$ being the total number density, $r_{med}$ the median radius and $\sigma$ the mode width of the particle size distribution. These parameters needed to be retrieved.

The aerosols are assumed to be composed of a solution of 75% $H_2SO_4$ and 25% $H_2O$. Furthermore, stratospheric aerosols are assumed to be spherical in shape, thus Mie theory can be applied. The possible values of the parameters to be retrieved are assumed to lie between 1 and $1000\,nm$ for the median radius and 1.05 and 2.0 for the mode width.



Before their use in the retrieval process, the extinction coefficients provided in the SAGE III/ISS data set were smoothed over altitude using a 1-2-1 smoothing scheme, aligning the version 5.21 data set with the previous 5.1 version, where a similar smoothing of the data was inherent. Extinction ratios at three wavelengths (449 nm to 756 nm and 1544 nm to 756 nm) are used to retrieve the median radius and the mode width. Using a Mie Code (Oxford, 2018) these extinction ratios were calculated for all combinations of a median radius value between 1 and 1000 nm in 1 nm increments and a mode width value between 1.05 and 2.0 in steps of 0.1. The real parts of the refractive indices at the used wavelengths that were necessary for the calculations were taken from Palmer and Williams (1975) and Lorentz-Lorenz-corrections, as described by Steele and Hamill (1981) were applied to them to obtain refractive indices at typical lower stratospheric temperatures. With these Mie calculations a kind of two-dimensional lookup-table with known median radius and mode width values for a given combination of extinction ratios is calculated. Forming extinction ratios at the same wavelengths from the SAGE III/ISS extinction coefficient data set, both size distribution parameters can then be retrieved by means of interpolation using the lookup-table. More details are provided in Wrana et al. (2021).

The effective radius which is used for the comparison to the model simulations in Sect. 6 is calculated from median radius and mode width with the following relation:

$$r_{eff} = r_{med} \cdot exp(\frac{5}{2} \cdot ln^2(\sigma)) \tag{2}$$

Also, an accuracy parameter $a$ as defined by Wrana et al. (2021), which is calculated from the distance between the curves of the lookup table and the error bars of the extinction ratios calculated from the SAGE III/ISS extinction coefficients, is used to exclude noisy data in the retrieved quantities shown in this work.

## 2.3 ECHAM model

The simulation of the Raikoke and Ulawun volcanic eruptions (see Sect. 6) were performed using MAECHAM5-HAM. ECHAM5 (Giorgetta et al., 2006), a general circulation model, was used in its middle atmosphere (MA) version, with a horizontal resolution of about 1.8°. It has a spectral truncation at wave number 63 (T63), with 95 vertical layers up to 0.01 hPa (about 80 km). To achieve realistic wind and transport conditions a nudging of the large wave numbers of the model to ERA5 reanalysis data was performed (Hersbach et al., 2018).

The aerosol microphysical model HAM (Stier et al., 2005) is interactively coupled to ECHAM. HAM includes the simulation of the oxidation of sulfur and sulfate aerosol formation, including nucleation, accumulation, condensation, and coagulation processes. Above the tropopause, a simple stratospheric sulfur chemistry was applied (Timmreck, 2001; Hommel et al., 2011) using prescribed oxidant fields of $OH$, $NO_2$, and $O_3$. In the simulations, the sulfate aerosols are radiatively active for both short wave and long wave radiation and coupled to the radiation scheme of ECHAM. The model setup and the setup of the



mode width is described in Niemeier et al. (2009) and Niemeier and Timmreck (2015), respectively. The parametrization of nucleation processes has been updated according to Määttänen et al. (2018), which slightly increased particle nucleation in the model.

## 2.4    TROPOMI instrument

For the comparison of model to observations data in Sect. 6 the emitted $SO_2$ masses of the 2019 Raikoke and Ulawun eruptions were estimated from measurements of the TROPOMI instrument onboard of the Sentinel-5 Precursor satellite. The data product used in this study assumes the $SO_2$ profile as a 1 km thick box filled with $SO_2$ and centered at 15 km altitude. We defined a grid with a latitude/longitude resolution of $0.1°$ times $0.1°$ from 30°N to 30°S and 110°E to 100°W. For the June and August 2019 eruptions of Ulawun, only $SO_2$ total vertical column data with a solar zenith angle less than 70° (Theys et al., 2022) and

with values less than $1000 \frac{mol}{m^2}$ were used. The vertical columns were multiplied by the $SO_2$ molar mass in order to obtain an $SO_2$ mass loading in units of $\frac{g}{m^2}$. Different thresholds of either $0 \frac{g}{m^2}$ or $0.05 \frac{g}{m^2}$ were applied to distinguish the volcanic signal from the background. The data was averaged for each grid segment and the $SO_2$ mass in units of grams was determined for each segment. Since some orbits spatially overlap, 14 consecutive orbits were bundled into a batch that covered approximately 24 h and averaged for each grid segment of the batch. The $SO_2$ mass in all grid segments of a batch were summed up to finally

receive the total $SO_2$ mass per batch. Depending on the threshold, the $SO_2$ mass for the June 2019 eruption was estimated as 0.12-0.16 Tg and for the August eruption as 0.18-0.2 Tg. The $SO_2$ mass emitted by the Raikoke eruption in June 2019 was calculated using no threshold with no restriction on the solar zenith angle, but with the requirement that the quality value needs to be larger than 0.5. This is described in more detail in Muser et al. (2020) and resulted in an $SO_2$ mass estimate of 1.37 Tg.

## 3    SAGE III/ISS timeline

In this section an overview of the volcanic eruptions investigated in this work is given. Three main periods of volcanic activity will be looked at: The Ambae eruptions of 2018, the Raikoke and Ulawun eruptions of 2019 and the La Soufrière eruption of 2021. In Table 1 summarizes information on the most important eruptions and eruptive phases in that time frame, important insofar as the eruptions were explosive enough to inject $SO_2$ directly into the stratosphere.

         Figure 1 gives an overview of the evolution of the extinction coefficient at 449 nm from the SAGE III/ISS data in the tropics

to give a sense of the temporal order of the volcanic eruptions investigated in this work. Daily zonal averages between 30°S and 30°N are shown for the time between 2018 and early 2022. Darker colors indicate higher values. The panel above shows which latitude the measurements of the averaged profiles beneath correspond to, due to the latitudinal shift of the SAGE III/ISS measurements. Red dashed lines indicate the dates of volcanic eruptions, whose signatures can be seen as darker colors in the contour plot. Since the latitudes of the SAGE III/ISS occultation measurements change from day to day, there are gaps in the

sampling of any particular latitude band such as the tropics in this case, which explains the time gaps in the color plots.





| | Latitude | Longitude | Date | SO$_2$ emission estimate |
|---|---|---|---|---|
| Ambae 1 | 15°S | 168°E | March – April 2018 | 0.1 Tg |
| Ambae 2 | | | July 2018 | 0.4 Tg |
| Raikoke | 48°N | 153°E | June, 21$^{st}$/22$^{nd}$ 2019 | 1.37 Tg |
| Ulawun 1 | 5°S | 151°E | June, 26$^{th}$ 2019 | 0.14 Tg |
| Ulawun 2 | | | August, 3$^{rd}$ 2019 | 0.3 Tg |
| La Soufrière | 13°N | 61°W | April, 9$^{th}$ – 22$^{nd}$ 2021 | 0.4 Tg |

**Table 1.** Information related to the most important volcanic eruptions and eruptive phases leading to direct injections of SO$_2$ into the lower stratosphere in the time period discussed in this work and covered by the SAGE III/ISS instrument, i.e. 2018 – 2021. Sources: (Muser et al., 2020; Kloss et al., 2020, 2021; Joseph et al., 2022; Bruckert et al., 2023)

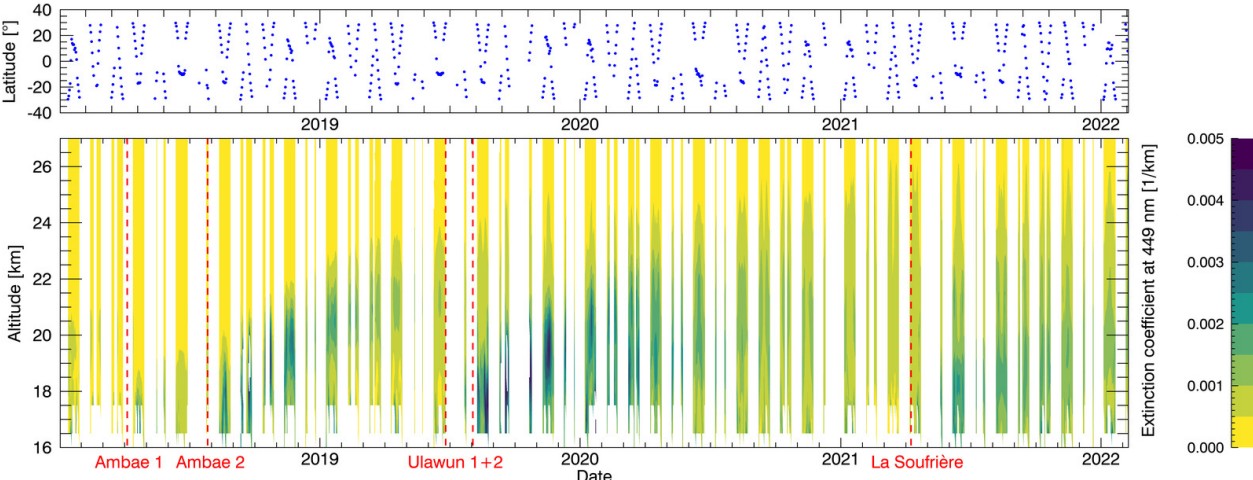

**Figure 1.** Bottom panel: Daily zonal means of the extinction coefficient at 449 nm between 30°S and 30°N. Top panel: Geolocations of the SAGE III/ISS solar occultation measurements used. Dashed vertical lines indicate volcanic eruptions that happened during the depicted time frame and reached the stratosphere.

We observe that aerosol extinction coefficient at 449 nm is around $10^{-4}$ km$^{-1}$ in the main aerosol layer from the start of the mission until the April 2018 eruption of Ambae. As would be expected, extinction increases in the lower stratosphere after each individual volcanic eruption. As described by Vernier et al. (2011), the new aerosol in the lower tropical stratosphere rises slowly to higher altitudes in a manner that mimics the water vapor "tape recorder".

Similarly to Fig. 1, Fig. 2 shows the extinction coefficient at 449 nm but for latitudes between 35°N and 70°N. Due to the lower tropopause height here, which also results in the stratospheric aerosol layer residing at lower alitudes, a different altitude window is shown. Notice also the difference in color scale value range. Here, the signature of the Raikoke eruption, which was



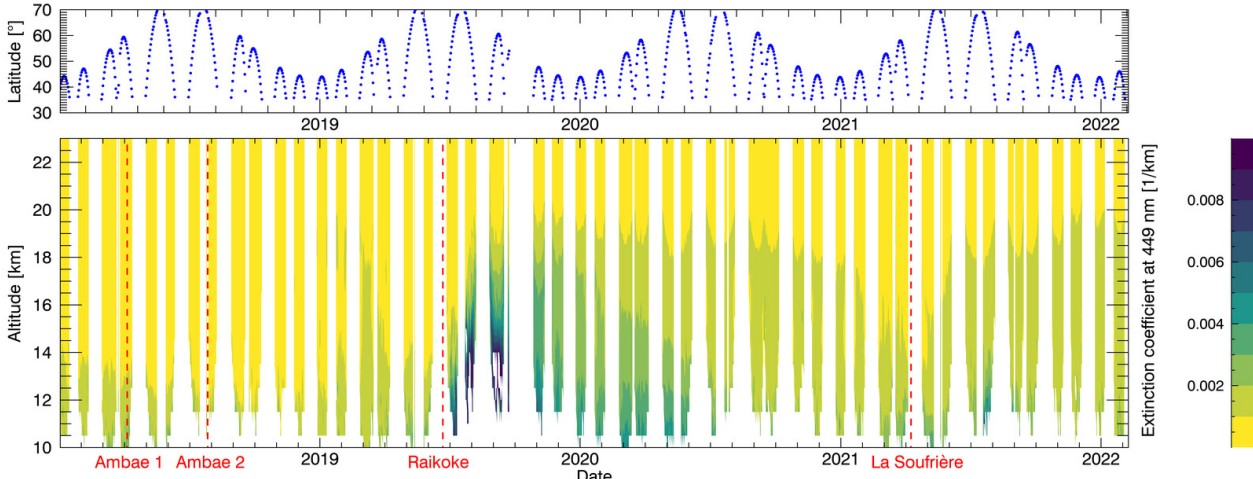

**Figure 2.** As Fig. 1 but for the northern hemisphere between 35°N and 70°N. Notice the different altitude and color scale ranges.

the strongest of the eruptions discussed in this work, dominates. It has to be noted, that in this region, as well as in Fig. 1, the signatures by the Raikoke and Ulawun eruptions cannot be clearly separated from each other. Although the perturbations are
much lower it can be seen, that the aerosol plumes of the Ambae and La Soufrière eruptions also reach the higher northern latitudes. After the Ambae eruptions this takes several months, which is most likely because the Ambae volcano is located in the southern hemisphere (15°S), but may also have to do with the stronger second eruption happening in July, when transport out of the tropics into the northern hemisphere is blocked by the subtropical transport barriers (Shuckburgh et al., 2001; Niemeier and Schmidt, 2017). The lower bound of the individual daily averaged profiles within a month as well as the extinction signals
visibly correlate with latitude, which is again due to the tropopause height strongly varying with latitude in this latitude range.

## 4 SAGE III/ISS monthly zonal means

In this section, the temporal and spatial evolution of different quantities related to the stratospheric aerosol size distribution during the three major phases of volcanic activity that are visible in Figs. 1 and 2 is discussed. Figure 3 relates to the months around the Ambae eruptions in 2018, Fig. 4 to the Raikoke and Ulawun eruptions in 2019 and Fig. 5 to the La Soufrière
eruption in 2021. Within each of these figures a row corresponds to a certain month. Note that only a selection of months is shown, which are chosen to give a good overview of the evolution of the particle size distribution from before the start of the eruption/s (top row) to the waning of the observed signals. Besides saving space this is also done because some of the months not shown here contain large data gaps for broad latitude bands due to the orbit of the ISS. Each column depicts a different parameter. Those are, from left to right, the median radius, the absolute mode width (see below), the number density and the
extinction coefficient at 449 nm. The mode width $\sigma$ and the effective radius are shown in the Appendix, since both quantities are not essential for the understanding of the observed effects. Each individual plot within Figs. 3 to 5 contains monthly zonal





means for 5° latitude bins. The red line indicates the tropopause height as provided in the SAGE III/ISS data set. All plots of the same parameter are shown with the same value range to enable an easy overview of its temporal evolution. Median radius and absolute mode width are plotted linarly, while number density and extinction coefficient are plotted logarithmically, due to the wide value range of the latter two parameters. To make this difference more apparent, two different color schemes are used, but in both of them lighter colors correspond to higher values. Only data points above the tropopause are shown. The latitudinal locations of the volcanoes relevant for the particular volcanic period are marked with triangles on the bottom of each plot.

It should be noted, that within each plot, profiles of different latitude bins correspond to different days within the month depicted. This is because of the latitudinal shift of the SAGE III/ISS sunrise and sunset measurements, which in turn is a result of the solar occultation geometry combined with the orbit parameters of the ISS.

The absolute mode width $\omega$ (second column in Figs. 3 to 5) is the standard deviation of the monomodal log-normal distribution in linear space, as introduced by Malinina et al. (2018). It is calculated from the median radius and mode width in the following way:

$$\omega = \sqrt{r_{med}^2 \cdot exp(ln^2(\sigma)) \cdot (exp(ln^2(\sigma)) - 1)} \qquad (3)$$

The absolute mode width $\omega$ is shown here instead of the mode width $\sigma$ because it is easier to interpret and because it provides direct information on the shape of the size distribution. In contrast, $\sigma$ can not be interpreted independently from the median radius in radius space and is, therefore, much less useful to understand how broad the particle size distribution actually is.

## 4.1 Ambae eruptions of 2017/2018

In Fig. 3, the effects of the Ambae eruptions on the stratospheric aerosol layer can be seen.

In the first row of plots the closest to background conditions before the two main eruptive phases in April and July 2018 are shown, namely monthly averages for March. Nevertheless, the number density and the extinction coefficient are already enhanced. For the northern hemisphere this can be attributed primarily to the Canadian wildfires of 2017, which was the burning season with the largest area burnt in British Columbia since beginning of the recording (Ansmann et al., 2018). However, there is also a signal in the southern hemisphere with enhanced number densities and extinction coefficients and low median radius values. This signal first showed up in the southern hemisphere in October 2017 already (not shown). This is more of a mystery but could be a consequence of several smaller tropical volcanic eruptions in late 2017, which, to our knowledge, did not directly inject $SO_2$ above the tropopause. These include the Tinakula (10.4°S) eruption on October $21^{st}$ whose ash and gas plume reached up to 10.7 km altitude (Laiolo et al., 2018), the Agung (10.4°S) eruption on November $21^{th}$ with an ash plume reaching 9.1 km (Global Volcanism Program, 2023) as well as Ambae's (15°S) first two, less explosive eruptive phases, which lasted from September $22^{nd}$ to October $6^{th}$ 2017 and from October $21^{st}$ to December $7^{th}$ 2017 (Moussalam et al., 2019). Since



these are all tropical volcanoes, it is conceivable, that some of the sulfur precursor gases emitted could have been transported into the lower stratosphere across the tropical tropopause layer (TTL) (Kremser et al., 2016).

210 The second row of plots in Fig. 3 depicts the month of June. Here, the effects of the third eruptive phase can be seen, which lasted from mid-March to mid-April 2018. This third eruptive phase was explosive enough to inject $SO_2$ directly into the lower stratosphere, with the largest injection period occurring on April, $6^{th}$. In the third row, which shows monthly means for September 2018, the impacts of the fourth and most active eruptive period of Ambae in July are visible. In July the largest amount of $SO_2$ was emitted, with 0.35–0.4 Tg $SO_2$. (Global Volcanism Program, 2023)

215 The extinction coefficient and the number density both show a distinct enhanced layer in the lowermost stratosphere above the tropopause that stands out from the much lower values at higher altitudes, where the signals of the Canadian wildfires and the unknown perturbation in the southern hemisphere in 2017 still relax more towards background conditions. This enhanced layer can be seen in June after the third eruptive phase but it becomes much stronger after the July eruptions. As described before, both hemispheres are affected by the eruptions (Malinina et al., 2021).

220 The most notable and surprising signal, though, is the strong decrease in median radius and absolute mode width in that same enhanced layer above the tropopause, mostly below 20 km altitude. This means, that the particle size distribution (PSD) shifts towards smaller aerosol radii, while also becoming more narrow. Therefore the Ambae eruptions seems to have led to a domination of a large number of smaller aerosol droplets in the lowermost stratosphere. It is also remarkable that the PSD stays in this configuration of on average very small aerosol particles for many months, as in January the signal is still very clear, 225 although already diminished. These findings will be discussed more in depth in Sect. 5. Over time, an interesting layering emerges in the tropics that is visible best in January 2019 (lowermost row), where a layer of larger median radius and absolute mode width values resides at roughly 21 km altitude with very small average aerosol sizes above and below.

### 4.2 Raikoke and Ulawun eruptions 2019

In Fig. 4 the PSD parameter evolution for the eruptions of Raikoke and Ulawun in the summer of 2019 are depicted. In the 230 first row, an average over the time between June $1^{st}$ to $20^{th}$ is shown, i.e. before the Raikoke and Ulawun eruptions on June $22^{nd}$ and June $26^{th}$, respectively. This is the closest to background conditions before this phase of volcanic activity. At this point, the effects of the Ambae eruptions in the year before have diminished strongly, although the extinction coefficient is still slightly enhanced throughout the main Junge layer.

In August, after the June eruptions of Raikoke and Ulawun and after the second Ulawun eruption on August $3^{rd}$, an increase 235 in the extinction coefficient over both the northern and southern hemisphere occurred, as expected. The signal is much stronger in the northern hemisphere, since the Raikoke eruption emitted more $SO_2$ (around 1.37 Tg) than both Ulawun eruptions together (around 0.44 Tg combined).

However, regarding the PSD parameters a unique pattern emerges, where Ulawun and Raikoke had opposite effects. Over the broad Raikoke area and in the lowermost stratosphere below roughly 15 km the median radius, absolute mode width 240 and number density increase from June to August. In contrast, in the southern hemisphere median radius and absolute mode width values show a strong decrease. This goes along with an especially strong increase in number density in the southern







**Figure 3.** Median radius (leftmost column), absolute mode width $\omega$ (second column), number density (third column) and extinction coefficient at 449 nm (rightmost column) for characteristic months before and after the Ambae eruptions in 2018. The location of the volcano is marked with a triangle and letter on the bottom of each plot and tropopause height is indicated by a red line.





hemisphere. This means that the southern hemisphere is dominated by a high number of very small aerosol particles below roughly 20 km, which notably stay small until the end of the year, similar to the effects of the Ambae eruptions in the previous year. The average aerosol size stays small at least until November 2019. In January 2020 the signal is overwritten by emissions stemming from the Australian wildfires of 2019/2020, which where unprecedented in the destruction caused and in the area affected by high-severity fire (Collins et al., 2021). The PSD parameters affected by these wildfires in the last row of plots are not reliable though, since the assumptions on the shape of the size distribution as well as on the refractive indices of the aerosols made in the retrieval of this work may not be realistic for these conditions.

Although the Raikoke eruption emitted around 15 Tg of ash (Osborne et al., 2022), it was only detectable in the atmosphere for a few days and roughly 90 % of it was removed from the atmosphere within 48 hours (Prata et al., 2022). Therefore, it is unlikely to play a big role in the retrieval data presented in Fig. 4. Nevertheless, the signals visible in the northern hemisphere may not be attributable to the Raikoke eruption alone, since there also have been two different severe wildfire events on the northern hemisphere in 2019 that were strong enough to reach the stratosphere (Voronova et la., 2020; Osborne et al., 2022). It is still not clear if and to what extent there was interaction between smoke and the sulfate aerosol plume originating from the Raikoke eruption. Therefore it is not out of the question whether smoke may have played a role in the aerosol size increase retrieved in the volcanic period displayed in the northern hemisphere.

### 4.3 La Soufrière eruption 2021

In Fig. 5, the period of volcanic activity before and after the La Soufrière eruptions between April $9^{th}$ – $22^{nd}$ 2021 is depicted.

The first explosive eruptions happened on April $9^{th}$ throughout the day with the ash plume reaching up to 8 km. Explosive activity on April $10^{th}$ and $11^{th}$ led to ash plumes rising up to 16 km in altitude. In the following days more volcanic activity was observed, although plume heights did not exceed 12 km anymore (Yue et al., 2022; Bruckert et al., 2023). Based on TROPOMI measurements, La Soufrière emitted roughly 0.4 Tg of $SO_2$.

The patterns in the PSD parameters as well as in the extinction coefficient that emerge are very similar to those observed for the period of the Ambae eruptions in 2018 (Fig. 3). The aerosol cloud spreads over both hemispheres, visible in an increase in number density and extinction coefficient in the lowermost stratosphere slowly affecting higher altitudes over time. Notably, there again is a strong decrease in median radii and absolute mode widths after the La Soufrière eruptions in both hemispheres in the same altitude region. Both parameters stay very low due to the eruptions at least until January 2022. A slow rising of the enhanced aerosol layer in the tropics from around 20 km altitude in June 2021 up to around 23 km in November 2021 can be observed in all PSD parameters. On January $15^{th}$ the Hunga Tonga - Hunga Ha'apai eruption happened, which was very unusual in many ways and introduced perturbations into the stratosphere which are not shown or discussed here (Legras et al., 2022).





**Figure 4.** Analogous to Fig. 3 but for the months around the Raikoke and Ulawun eruptions.







**Figure 5.** Analogous to Fig. 3 but for the months around the La Soufriere eruption.



## 5   Discussion of the SAGE III/ISS retrieval results

This satellite observation of the size reduction in the aftermath of the Ambae, Ulawun and La Soufrière eruptions is unprecedented. While there is some evidence in previously published literature that some volcanic eruptions may lead to changes in
extinction ratios at two wavelengths (Rieger et al., 2014; Thomason et al., 2021), this cannot easily be interpreted as an increase or decrease of aerosol size, as will be shown below. Indeed, this kind of analysis using two wavelengths can be helpful to detect differences in the effect of volcanic eruptions on the aerosol size in the first place, but the amount of concrete information on the particle size distribution itself is inherently limited, since with only two pieces of independent spectral information the mode width is usually fixed at an assumed value in order to retrieve the median radius. This is a large source of error, since
the retrieval carried out in this work using the SAGE III/ISS data set strongly suggests that the mode width of a monomodal lognormal size distribution of stratospheric aerosol varies depending on time and space. As Malinina et al. (2019) pointed out, there are a lot of different possible combinations of median radius and mode width values that would lead to the same extinction ratio at two wavelengths.

This is illustrated in Fig. 6a, where 9 different calculated monomodal log-normal size distributions are shown, that are all
consistent with the same extinction ratio at two wavelengths, namely a ratio of 2.0 for the wavelengths 449 nm / 756 nm. Each of these PSDs is based on the retrieval of the median radius assuming a different mode width between 1.2 and 2.0 in steps of 0.1 using the extinction ratio of 2.0 with a standard two wavelength retrieval method. After the retrieval of the median radius the number density was calculated. The PSD plotted in blue is the result for the maximum assumed mode width of 2.0, which resulted in a median radius of 55 nm (corresponding to an effective radius of 183 nm), the red curve corresponds to the smallest
assumed $\sigma$ of 1.2 with a retrieved median radius of 253 nm (effective radius = 275 nm). These could all be valid solutions were the stratospheric aerosol size to be retrieved from measurements at only two wavelengths, i.e. we could not conclude in which way the size distribution of the stratospheric aerosol changes after volcanic eruptions. To emphasize this, note that the median radius values in this example almost span the range of the median radius variability observed in the SAGE III/ISS retrieval data of this work. In panel (b) of Fig. 6, the curves of the lookup-table that was used to retrieve the median radii of the PSDs
in panel (a) are depicted. For each curve extinction ratios at the wavelengths 449 nm and 756 nm were calculated using Mie theory for a range of median radius values. Each curve corresponds to a $\sigma$ value between 1.2 and 2.0. A different median radius value will be retrieved depending on the assumption of the mode width, i.e. no unique solution exists, as illustrated by the triangles marking the position of the examplary extinction ratio of 2.0. In contrast, using the second lookup-table shown in panel (c) (similar to what was used in this work), the mode width $\sigma$ can be retrieved along with the median radius, since
here, two extinction ratios using three wavelengths are used. These curves are calculated in the same way as the ones of panel (b), except that a third wavelength, in this case 1544 nm is included. If such a lookup-table is used, the triangles, although still marking the extinction ratio at 449 and 756 nm of 2.0, do not mark equally valid solutions for the particle size retrieval. Instead, due to the second extinction ratio, the mode width can be retrieved along with the median radius and does not have to be assumed anymore. This way, a unique solution emerges for both parameters and the ambiguity illustrated by the large
differences between the size distributions of panel (a) is eliminated.



This is where the observational data set presented in this work can contribute to our understanding of how the stratospheric aerosol size changes after volcanic eruptions. Since the retrieval method is based on three wavelengths (Wrana et al., 2021), much more information on the actual shape of the size distribution is gained, still under the assumption of a monomodal log-normal PSD.

On a side note, Fig. 6 also illustrates, why the mode width $\sigma$ is not a useful quantity to get a sense of how broad a PSD actually is. The blue size distribution with a $\sigma$ of 2.0 does not appear to be broader than the red curve with a $\sigma$ of 1.2. This is why the absolute mode width was shown in this work instead.

While the small sample size of volcanic eruptions investigated in this work makes it difficult to draw generalized conclusions, the volcanic eruptions of Ambae in 2018, Ulawun in 2019 and La Soufrière in 2021, which produced the strong reduction in 315 average stratospheric aerosol size, share some similarities. They emitted similar amounts of $SO_2$, i.e. close to $0.4\,\mathrm{Tg}$. In contrast, Raikoke, whose eruption in 2019 did not lead to an observed size decrease but instead to an increase, emitted around $1.37\,\mathrm{Tg}\ SO_2$. In addition, the former three volcanoes are all tropical, whereas Raikoke is situated in mid-northern latitudes, at around $48°N$. This could be important insofar as, depending on season, there are temperature differences in the lowermost stratosphere between low and mid latitudes. Temperature is an important factor in nucleation and condensation rates, which 320 are integral lifetime processes of stratospheric aerosol and play an important role in their size evolution (Kremser et al., 2016). Nucleation strongly increases with decreasing temperatures (Vehkamäki et al., 2002; Korhonen et al. , 2003) which can shift the sulfate aerosol size distribution towards smaller sizes (Pirjola et al., 1999).

The assumption that the true size distribution of stratospheric aerosols can be described well by a monomodal lognormal size distribution is at the core of the observational data presented here. This assumption is made frequently for satellite retrieval 325 data sets, firstly because it is a reasonable assumption for many cases, but also because it is often necessary to limit the number of unknown variables of the PSD in order to still be able to solve for them using the limited amount of independent information contained within a measurement. Indeed, in some cases the truth is probably closer to a bimodal lognormal distribution (Deshler et al., 2003). However, the false assumption of a monomodal log-normal size distribution in a bimodal log-normal case would lead to an overestimation of the particle size in a satellite occultation measurement data set like the SAGE III/ISS data used in 330 this work (von Savigny et al., 2020b). This is because the second mode, although containing far fewer aerosols, would contain larger aerosol particles which, in the size regime typical for stratospheric aerosol, are much more efficient scatterers and would dominate the spectral signal picked up by the instrument. In turn, a retrieval based on the assumption of a monomodal log-normal size distribution would produce a PSD shifted towards larger radii. Therefore, the signal of a size distribution shifted towards very small radii, that is observed in the volcanic periods discussed in this study, can not be the result of a wrong 335 assumption on the size distribution shape.



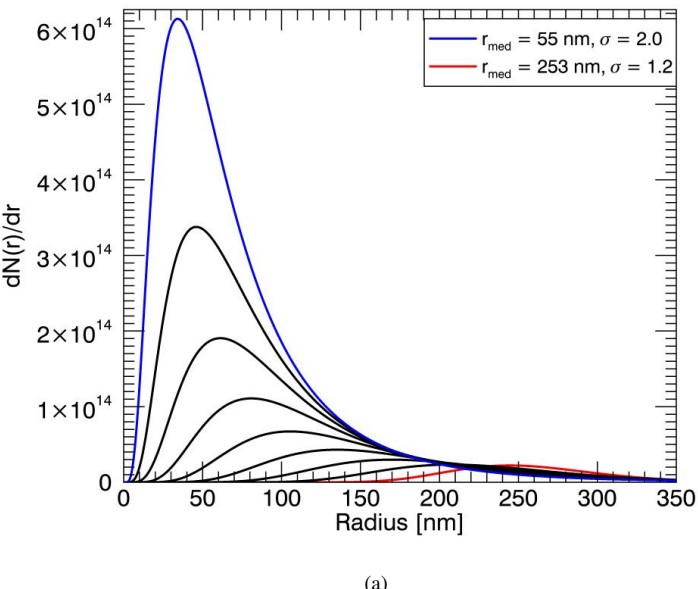

(a)

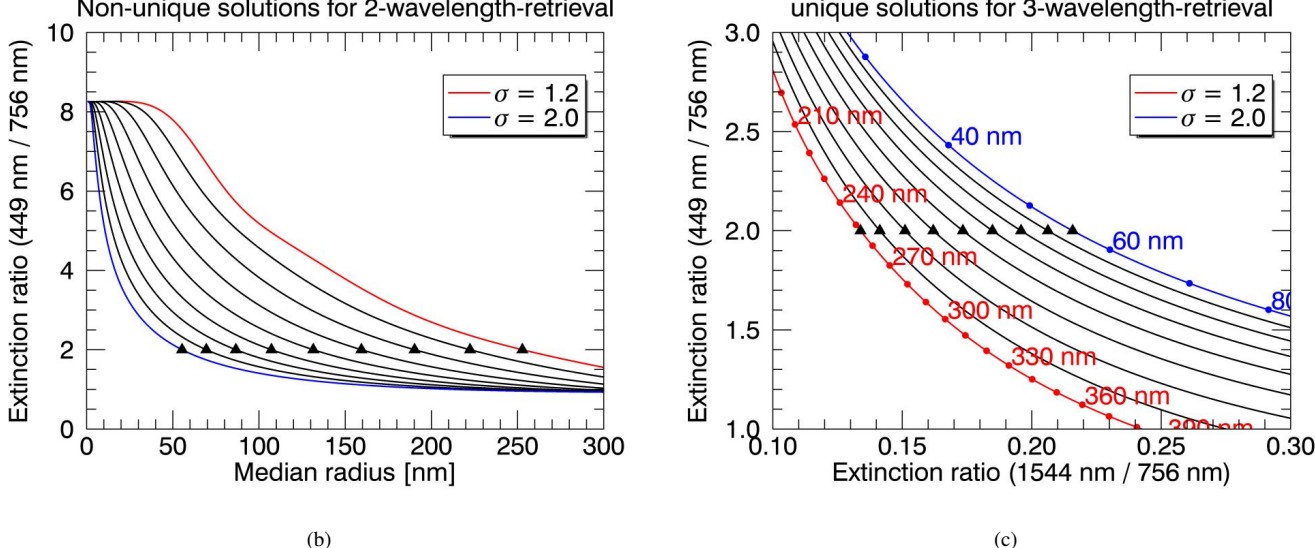

(b)

(c)

**Figure 6.** Panel (a) shows nine different exemplary monomodal log-normal size distributions with different values of the mode width $\sigma$ between 1.2 (red curve) and 2.0 (blue curve) in steps of 0.1. All PSDs depicted are consistent with an extinction ratio of 2 at the wavelengths 449 nm / 756 nm. Panel (b) shows the lookup-table used to retrieve the median radii of the PSDs in panel (a), using 2 wavelengths, with which the true size distribution can not be identified. Triangles indicate the different non-unique solutions. Panel (c) shows a lookup-table using 3 wavelengths and 2 extinction ratios, for which unique solutions exist, since each triangle corresponds to a different value of the second extinction ratio.





## 6 Comparison: Model vs Observations

In order to learn whether current models, in this case the ECHAM model, can help us to identify and understand the main dynamical and microphysical factors controlling the aerosol size evolution after volcanic eruptions, specifically the observed particle size reduction, model simulations of the time period before and after the Raikoke and Ulawun eruptions in 2019 were

carried out. This period of volcanic activity was chosen, because here an average aerosol size reduction is found in the southern hemisphere and tropics (over Ulawun) alongside an average aerosol size increase in the northern hemisphere (over Raikoke) at the same time. This makes it a useful case to test whether the model can reproduce the differences in aerosol size evolution after these eruptions in order for it to be used to gain a better understanding of the related processes.

### 6.1 Simulation of Raikoke and Ulawun 2019

In the simulations, $SO_2$ masses that are vertically resolved in three altitude levels are injected into the lower stratosphere at the locations of the Raikoke and Ulawun volcano at the time of each of the eruptions. Consequently, the evolution of the particle size distribution of the forming sulfate aerosols is calculated. Multiple model runs were performed with varying input parameters. The relevant parameters used for the best model run, which is used in this work, are shown in Table 2. The $SO_2$ mass injected into the stratosphere for the second eruption of Ulawun is taken from Kloss et al. (2021) and parameters for the

other two eruptions come from our own $SO_2$ mass estimation using the TROPOMI data set. Like in the SAGE III/ISS retrieval the stratospheric aerosols are assumed to be composed of sulfuric acid and water only. In order to start the Raikoke and Ulawun simulations with background conditions comparable to the observations the simulation is started from a 10 year simulation on January $1^{st}$ of 2018. By doing that, the Ambae eruptions are included in the run. If the Ambae eruptions were not included, the simulated atmosphere would end up too "clean" in June 2019, before the Raikoke and Ulawun eruptions, and therefore not

be comparable to observations.

| | Raikoke | Ulawun | |
|---|---|---|---|
| Latitude | 48°N | 5°S | |
| Longitude | 153°E | 151°E | |
| Date of eruption | 22.06.2019 | 26.06.2019 | 03.08.2019 |
| Injected SO2 mass | 1.37 Tg | 0.14 Tg | 0.3 Tg |
| Injection Pressure Level | 140 hPa | 100 hPa | 90 hPa |

**Table 2.** Relevant parameters of the Raikoke and Ulawun eruptions as used in the ECHAM simulations.

In the model simulations ash is included, but there is no interaction with the sulfate aerosols, i.e. no mixed ash-sulfate-aerosols are calculated. Regarding the aerosol particle size, ECHAM uses a modal model. Four log-normal distributions are implemented, which are called, sorted from smaller to larger radii: the nucleation mode, Aitken mode, accumulation mode and



coarse mode. Each of these modes has a fixed mode width $\sigma$ and a median radius that can change within certain value ranges.
The number densities within the individual modes change as well, as a result of the calculated microphysical processes like the nucleation, condensation and coagulation rates.

## 6.2    Extinction

Here, the extinction coefficient at 550 nm is compared between the observational SAGE III/ISS data and the ECHAM simulations. In the SAGE III/ISS solar occultation level 2 data set the extinction coefficient is not provided at 550 nm directly.
To adress this, a third order polynomial was fitted to the extinction spectra using the 6 most reliable wavelength channels of the data set, i.e. 449, 521, 756, 869, 1021 and 1544 nm, to retrieve extinction coefficients at 550 nm that can then be used for comparison.

In Fig. 7, zonal averages of the extinction coefficient at 550 nm are depicted for the time before the volcanic eruptions (left column), i.e. between $1^{st}$ and $20^{th}$ of June 2019, and for August of 2019 (right column). The panels in the upper row show
the data that was calculated from the SAGE III/ISS measurements, while the panels in the lower row depict results from the ECHAM simulations. As explained in Sect. 2, the spatial and temporal sampling of SAGE III/ISS is limited. Therefore, to aquire an actually comparable temporal and spatial coverage between the observational and model data set, we applied the sampling of the SAGE III/ISS measurements to the model output. In other words, only profiles from the ECHAM simulations for locations and times were included, where SAGE III/ISS measurements happened on the same day within 1 degree of
longitude and 2.5 degrees of latitude. As in Sect. 4 this means, that data in different latitude bins corresponds to different days within the month depicted. The tropopause height shown was provided in the SAGE III/ISS lvl 2 data set, taken from the MERRA-2 reanalysis data. The blue line marks a 1 km interval above this calculated tropopause height, above which it is unlikely that the measurements are affected by clouds or include tropospheric air in general.

The background conditions in prevolcanic June 2019 are reproduced very well by the ECHAM simulations, both in mag-
nitude of the values and in the spatial patterns emerging, aside from an enhanced extinction between 40°N and 60°N below 14 km in the SAGE III/ISS data. This perturbation may be smoke from the two larger wildfires that reached the lowermost stratosphere, which is not included in the model. For August 2019, in general the ECHAM simulations could reproduce the observations as well, although in the SAGE III/ISS data the perturbations of the extinction coefficient partly reach higher altitudes in the northern latitude. Both model and observations show a strong increase of the extinction, with a much stronger
effect over the latitudes near Raikoke.



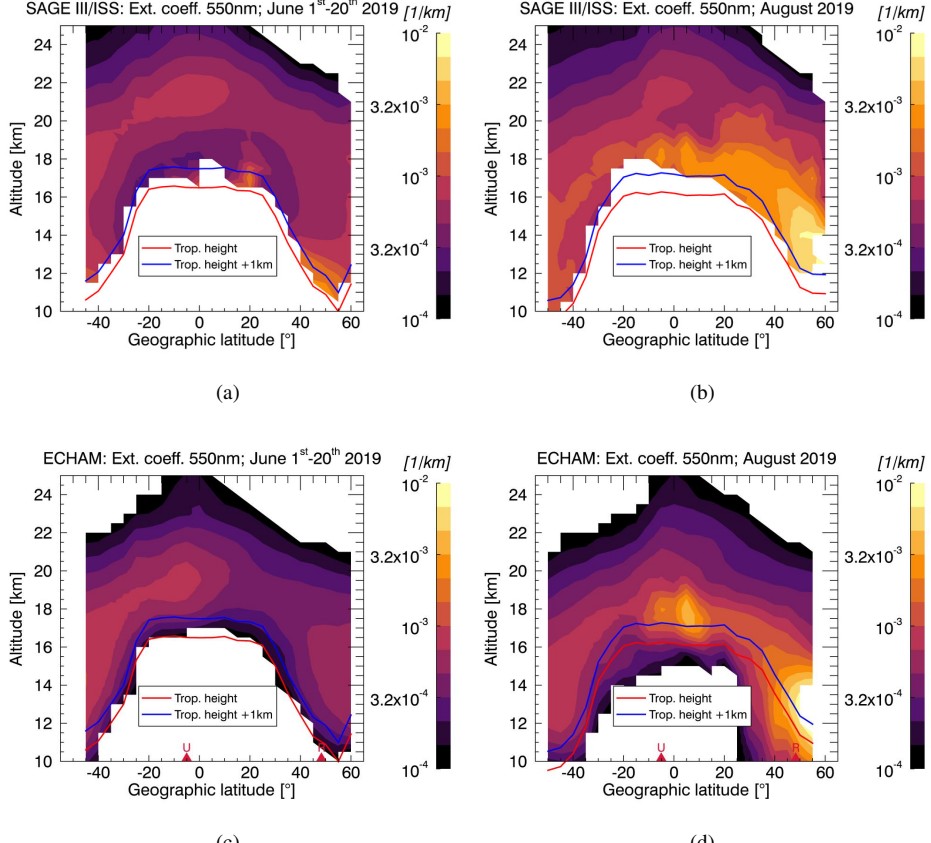

**Figure 7.** Zonal means of the aerosol extinction coefficient at 550 nm for June $1^{st}$ to $20^{th}$ (left) and August 2019 (right) from the SAGE III/ISS data set (upper plots) and from the ECHAM simulations using the spatial and temporal sampling of SAGE III/ISS (lower plots). Triangles on the bottom of each plot indicate the volcanoe's locations. Tropopause height is illustrated by a red line, with the blue line indicating an uncertainty interval of 1 km above tropopause height.

## 6.3 Effective Radius

The effective radius is the area-weighted mean radius of the size distribution (Grainger, 2017). It is a useful quantity since with it a particle size distribution can be expressed using only one parameter. Furthermore, it can be used to compare PSDs of different shape. This is necessary in this case, since our SAGE III/ISS retrieval data set includes monomodal log-normal

size distribution, while in the ECHAM model output, the PSD is expressed in terms of four individual log-normal modes. The effective radius $r_{eff}$ from the ECHAM simulations, representing these 4 modes, is calculated in the following way:

$$r_{eff} = \frac{m_3}{m_2} = \frac{V}{A} \cdot 3 \tag{4}$$





where $m_3$ is the third moment, $m_2$ is the second moment, $V$ is the the volume and $A$ is the surface area of the particle size distribution.


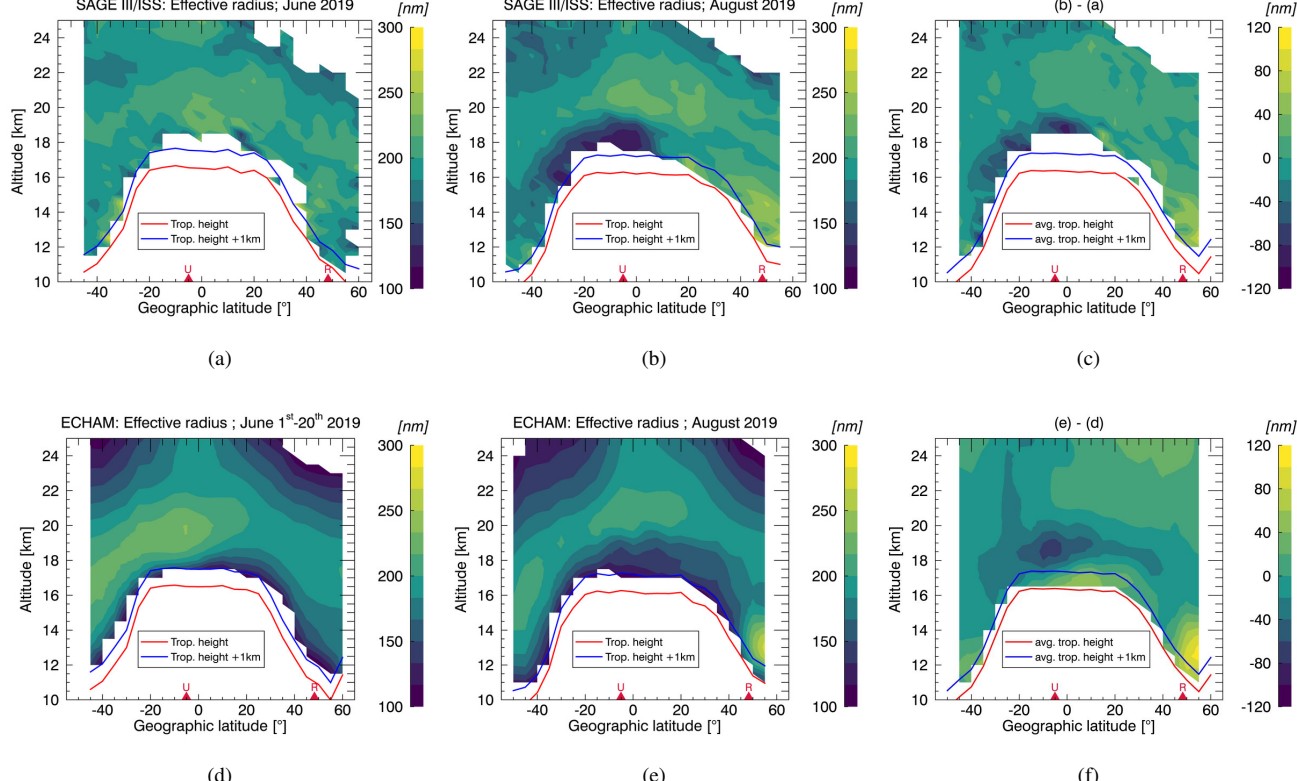

**Figure 8.** Zonal means of the effective radius retrieved from SAGE III/ISS (upper row) and calculated with the ECHAM model using the spatial and temporal sampling of SAGE III/ISS (lower row) for June $1^{st}$ to $20^{th}$ (leftmost column) and August 2019 (middle column). The rightmost column shows the temporal anomaly of the effective radius, i.e. the difference between the first two plots in each row. Triangles on the bottom of each plot indicate the volcanoe's locations. Tropopause height is illustrated by a red line, with the blue line indicating an uncertainty interval of 1 km above tropopause height.

The effective radii retrieved from the SAGE III/ISS observations in Figs. 8a and 8b, show an increase over the Raikoke area from June to August 2019 in the lowermost stratosphere roughly between 40 and 60°N. Most notably, there is a large reduction in the effective radius values over the Ulawun area (around 5°N to 35°S) over the same time span, analogous to the reduction in median radius and absolute mode that was discussed before in Sect.5. This reduction of average particle size persists at least

until November 2019 (see Appendix). In panel (c), the effective radius temporal anomaly, i.e. the difference between panels (b) and (a), is depicted, which makes it clear where the effective radius increased and where it decreased from June to August. Over the Raikoke area an increase of the effective radius by up to 87 nm is found and in the tropics and southern subtropics





there is a decrease by up to 123 nm. It has to be noted that part of the Raikoke plume has been missed because of the limited spatial coverage of the SAGE III/ISS measurements, since a substantial part of the plume was transported further north than
60°N (Kloss et al., 2021).

The lower row of panels in Fig. 8 is analogous to the upper row, but showing the ECHAM simulations for the same time frames. For the background conditions (panel (d)) effective radii in the lowermost kilometer above the tropopause are much lower than in the SAGE III/ISS retrieval data. Apart from this effective radii are close to the observations. In August, the qualitative pattern of high values in the lowermost stratosphere over Raikoke and low values over the broad Ulawun region
is reproduced by the model. Also in panel (f), which again shows the temporal anomaly, the absolute increases and decreases in the regions affected by the volcanic eruptions mostly match the observations well. The effective radius increase over the Raikoke area in the lowermost stratosphere is stronger in the model, which is because of the low effective radii in this region in the model's background. The smaller effective radii in panel (e) and the region of decrease in panel (f) reach further into the northern hemisphere than in the observations.

Starting in September 2019, model and observations start to diverge: Effective radii stay very low over the tropics and southern subtropics in the SAGE III/ISS retrieval data until december of 2019, before the Australian wildfires destroy the signal in January 2020. In contrast, the model calculates strong particle growth in the months after August, with effective radii increasing far beyond the background at higher altitudes. This suggests, that the MAECHAM5-HAM model can be used to learn about the stratospheric aerosol evolution and its underlying mechanisms in the short-term after volcanic eruptions, but not
necessarily for longer periods. Identifying the cause of this discrepancy between model and observations is difficult. Possible causes could in principal be an overestimation of coagulation by the model, deviations in dynamics, e.g. due to smaller vertical advection in ECHAM compared to other models (Niemeier et al., 2020), or biases in the observational data set or the retrieval algorithm.

## 7 Conclusions

Using a multi-wavelength extinction ratio approach we retrieved particle size distribution parameters of stratospheric aerosol from the SAGE III/ISS satellite solar occultation data set. As a result of the assumption of a monomodal log-normal size distribution, we retrieved the median radius, mode width and total number density and for understandability and comparison purposes the absolute mode width and effective radius as well.

We looked at the temporal evolution of these parameters in three different periods of volcanic activity in the SAGE III/ISS
data set, the first one being associated with the eruptions of Ambae in 2018, the second one with the Raikoke and Ulawun eruptions in 2019 and the third one with the La Soufrière eruption of 2021. Surprisingly, we found that the average aerosol size decreased for all of the mentioned eruptions, except for the Raikoke eruption. This is very different from e.g. the Mt. Pinatubo eruption of 1991, which is probably the volcanic eruption on which the most research was done in total, where aerosol size increased (Deshler et al., 2003). For the Ambae, Ulawun and La Soufrière eruptions instead, the median radius, absolute mode
width and effective radius decreased strongly, i.e. the PSD became narrower and shifted towards smaller radii. We also showed,





that for this finding, the use of a three-wavelength-extinction approach to retrieving stratospheric aerosol size as opposed to a standard 2-wavelength-extinction approach was crucial. This way, the mode width did not have to be assumed and the strong ambiguity inherent to 2-wavelength retrievals is removed. Notably, this strong reduction of average aerosol size also lasted for months in each case and even over a year in La Soufrière's case, when the perturbed aerosol layer could evolve more or less
undisturbed.

In order to better understand the importance of different microphysical and dynamical processes at play in the size decrease as well as in the long lifetime of the very small average aerosol sizes, atmospheric models will be a necessary and important tool of investigation. Because of this we performed simulations of the Raikoke and Ulawun volcanic activity period in 2019 with the aerosol climate model MAECHAM5-HAM. To compare the monomodal log-normal size distributions of our SAGE III/ISS
retrieval data with the 4-mode log-normal PSD of the ECHAM model the effective radius was used. The model was able to well reproduce the spatial and temporal patterns that were observed in the SAGE III/ISS data in the extinction coefficient at $550\,\mathrm{nm}$ and the effective radius from June to August 2019, i.e. the decrease in average stratospheric aerosol size over Ulawun and the increase over Raikoke. In this study we found it to be essential to include preceding large injections of sulfur-containing gases into the stratosphere, in this case the eruptions of the Ambae volcano in 2018, since without it the simulated atmosphere would
be too clean in the background conditions before Raikoke and Ulawun compared to observations. Despite this encouraging agreement with the SAGE III/ISS retrieval data in the first two months, there is a growing discrepancy between model and observations in the months thereafter. In other words, the model could not reproduce the long lifetime of the small aerosol size and instead the effective radii strongly increased in the simulations after the initial PSD parameter reduction over Ulawun.

Further research is needed, especially on the conditions necessary for a stratospheric aerosol average size decrease after
volcanic eruptions to occur as opposed to a size increase. This may be an important pathway to improve the capability of climate models to reproduce observed effects of volcanism on climate and reduce uncertainty of projections, when including volcanic eruptions. For this we will need more intercomparison between observational data and model simulations in terms of stratospheric aerosol size.

## Appendix A: Other quantities related to aerosol size

Here, we provide two additional quantities related to the stratospheric aerosol size distributions retrieved from the SAGE III/ISS data set for the three main volcanic periods covered in Figs. 3 to 5. The two quantities are the effective radius, which is explained in Sect. 6.3 and the mode width $\sigma$ or geometric standard deviation, which is part of Eq. 1.



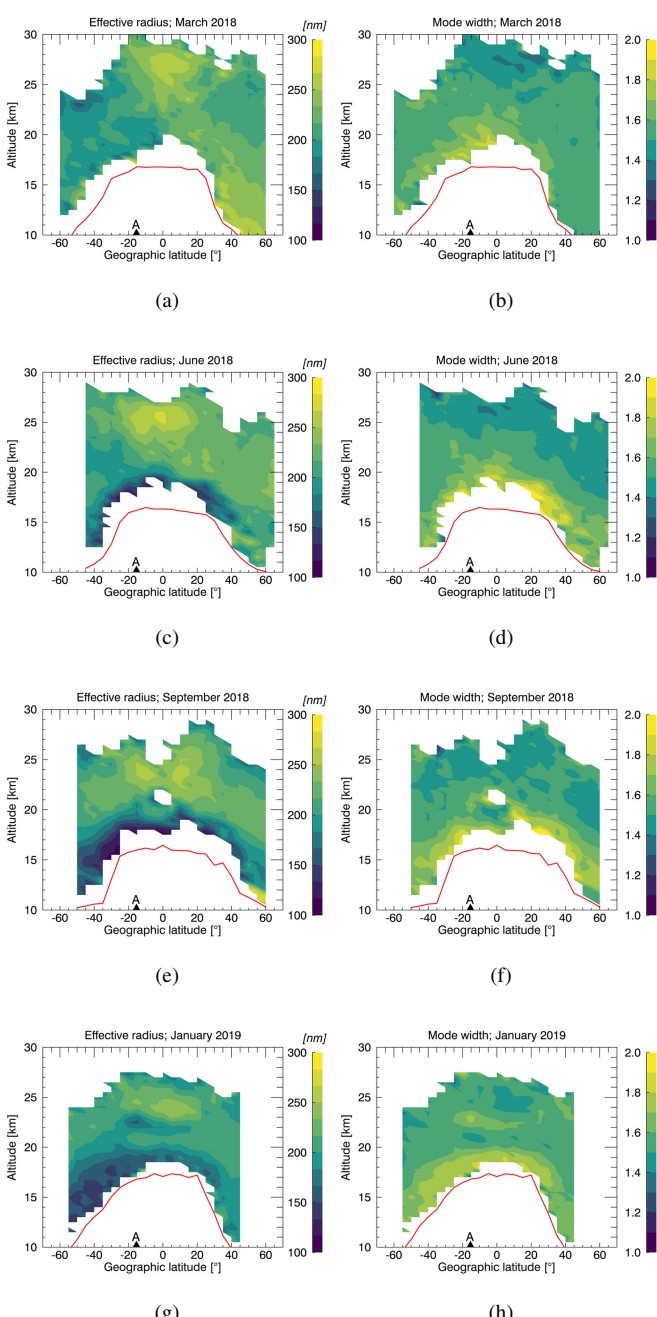

**Figure A1.** Analogous to Fig. 3, but showing the effective radius and the mode width $\sigma$ for the characteristic months before and after the Ambae eruptions in 2018.




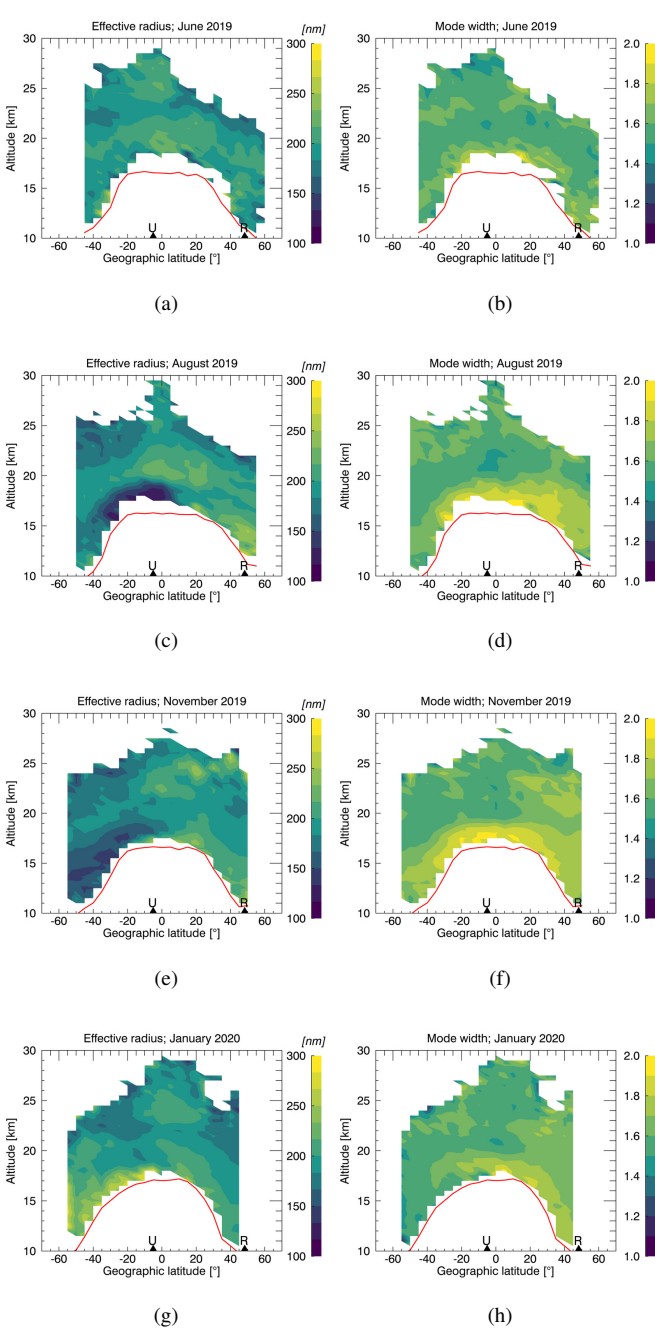

**Figure A2.** Analogous to Fig. A1, but for the characteristic months before and after the Raikoke and Ulawun eruptions in 2019.




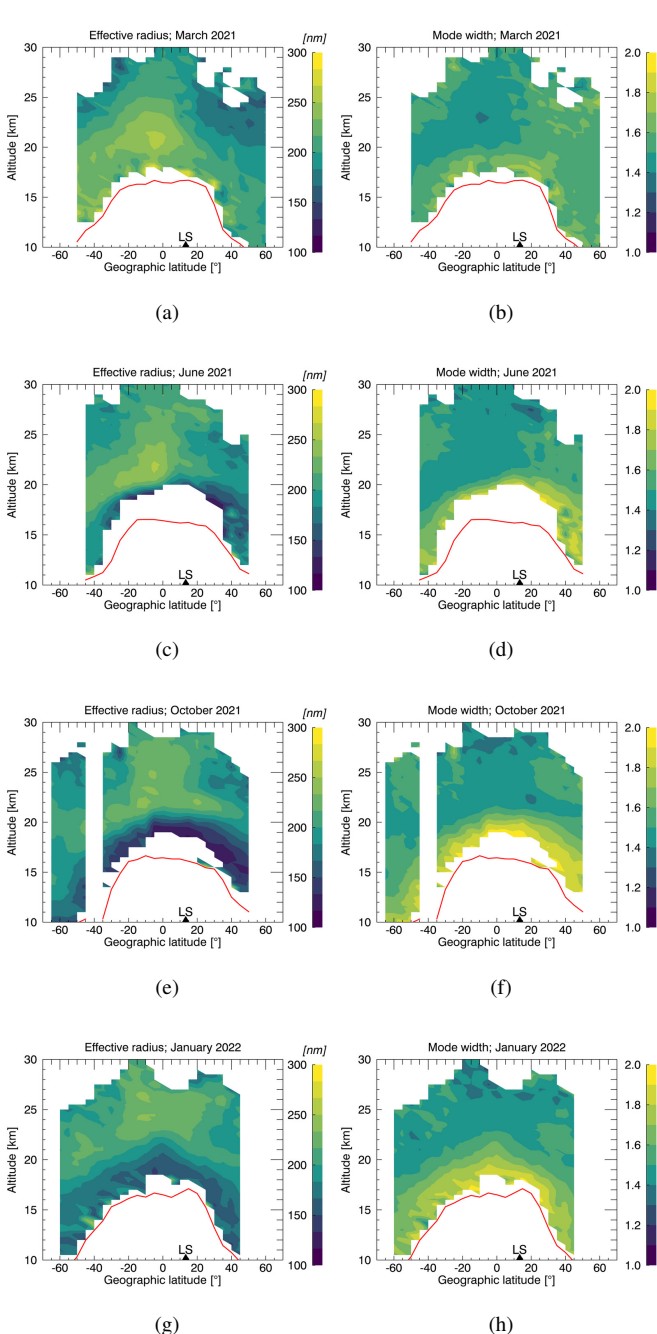

**Figure A3.** Analogous to Fig. A1, but for the characteristic months before and after the La Soufrière eruption in 2021.



*Data availability.* The data published in this manuscript can be obtained upon request to the first author. The SAGE III/ISS data was obtained from the NASA Earthdata Atmospheric Science Data Center (https://eosweb.larc.nasa.gov)

*Author contributions.* UN performed the simulations using the MAECHAM5-HAM model. FW performed the retrievals from the SAGE III/ISS data set and processed the ECHAM model data for comparison with the observational data. The findings were discussed by FW, CvS and UN. LWT provided insights into the SAGE III/ISS instrument and issues related to its measurements. SW estimated $SO_2$ masses emitted during volcanic eruptions using the TROPOMI data set. All authors discussed, edited and proofread the manuscript.

*Competing interests.* The authors declare that they have no conflict of interest.

*Acknowledgements.* This work was funded by the Deutsche Forschungsgemeinschaft (DFG, project VolARC (no. 398006378) of the DFG research unit VolImpact (FOR 2820)). We also acknowledge support by the University of Greifswald and thank the Earth Observation Data Group at the University of Oxford for providing the IDL Mie routines used in this study.



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
