# Peer review of "Stratospheric aerosol size reduction after volcanic eruptions"

_EGUsphere, 2023_

## Author Response (AR1)

**Anonymous reviewer #1 comments**

**Comment:** This is a useful contribution on determining stratospheric aerosol size distributions from SAGE III/ISS data and using the size distributions to assess the evolution of aerosol size following several recent small volcanic eruptions. The paper suffers some confusion over the definition and use of effective radius and in the presentations of distributions of dn/dr. Once these are clarified the paper should be published.

Here are detailed comments using the manuscript line numbers.

**Answer:** Thank you very much for the constructive and helpful criticism, especially of places, where we got the nomenclature wrong. We implemented most of the suggestions right away and made sections clearer, where there may have been a misunderstanding.

**Comment:** 81 Is σ the mode width or the distribution width? This is a monomodal lognormal so there is only one mode. Mode width is used throughout, but really what is intended is the distribution width, which is a better description.

**Answer/Changes:** We changed it to distribution width in the whole document.

**Comment:** 90 To be clear use : or / notation when writing the ratios. According to the text the ratios used are 449:756 and 1544:756. This seems a bit odd, one ratio is lesser:greater and the other greater:lesser? But this is what is done, Fig. 6, so the text is interpreted correctly as written.

**Answer/Changes:** We implemented your suggestion and now use ":" for the ratios. The choice of whether it is 1544:756 or 756:1544 does not affect the retrieval results. The orientation of the lookup-table would be different and subsequently the whole retrieval routine would have to be adapted to the changed order. Here, the ratio is simply described and shown the same way as in Wrana et al. (AMT, 2021), where the retrieval method was described in more detail.

**Comment:** 93-95 "The real parts of the refractive indices at the used wavelengths that were necessary for the calculations were taken from Palmer and Williams (1975) and Lorentz-Lorenz-corrections, as described by Steele and Hamill (1981) were applied to them to obtain refractive indices at typical lower stratospheric temperatures" is very confusing.

Try something like: The real parts of the refractive indices at the wavelengths used were calculated from Palmer and Williams (1975) using Lorentz-Lorenz-corrections. This has been described by Steele and Hamill (1981) to obtain refractive indices at typical lower stratospheric temperatures.

**Answer/Changes:** Thanks for the suggestion. We changed the phrasing to this:

"The real parts of the refractive indices at the wavelengths used were calculated from Palmer and Williams (1975) using Lorentz-Lorenz-corrections. These corrections have been described by Steele and Hamill (1981) and were necessary to obtain refractive indices at typical lower stratospheric temperatures."

**Comment:** 102 It should be mentioned here that this formula is just the result of the ratio of volume to surface area from the ratio of the third to second moments of a lognormal size distribution. The authors need to be clear for the readers that this expression does not include the factor 3 traditionally used in the definition of effective radius as V/A * 3, see Eqn. (4) in the discussion of the model results.

**Answer/Changes:** There seems to have been some confusion around the equations on the effective radius that we showed in the manuscript. There is no factor 3 difference between our satellite retrieval data and our model simulation data. Still, to avoid any further confusion we removed the "V/A * 3" from eq. (4) , since the effective radius from the ECHAM model simulations was calculated directly from the third and second moments of the size distributions. Eq. (2) should therefore be without potential for misunderstanding.

**Comment:** 189 It is unlikely that a majority of readers will be familiar with the absolute mode width. Perhaps it would be worthwhile to provide some examples of the relationship of omega with sigma for the range of median radii covered.

**Answer/Changes:** Thank you for the helpful suggestion. We added a figure to this section of the manuscript to make the relationship between omega and sigma easier to understand for the reader.

**Comment:** 2020 Delete already.

**Answer/Changes:** We deleted it.

**Comment:** 245 Change "where" to were.

**Answer/Changes:** Thank you for spotting the typo, we corrected it.

**Comment:** 274-275 Isn't it pretty well established that volcanic eruptions generally lead to changes in the extinction ratios between the longer and shorter wavelengths? In fact that is often the way that particle size following an eruptions is generally assessed. Thus "may" is not quite right here.

**Answer/Changes:** Correct, the word "may" was misplaced here. We deleted it.

**Comment:** Fig. 6a) Is something else being added to this figure besides just the differences in median radii and σ? Using the values in the legend of Fig. 6a), equation (1), and fiddling with No to roughly reproduce the y-axis in Fig. 6a), leads to the attached figure where the color coding, median radii, σs, and axes are the same as Fig. 6a). According to the figure caption only σ  is changed in steps of 0.1, but doesn't each change in sigma also require a change in the median radius. In fact this is explained in the text, but should also be included in the figure caption. Still it seems that the figure should be easily reproduced for the specific sizes and widths quoted, and yet the attempt shown here is not consistent with Fig. 6a).

**Answer/Changes:**

There has been a misunderstanding here, and we changed the text as well as the figure caption to make the following points clearer: The different sigma values are different values assumed for a PSD parameter retrieval from a hypothetically measured extinction ratio of 2 at the wavelengths 449nm and 756nm. For each different sigma value that is assumed the retrieval gives a different median radius value, so yes the median radius is different between the curves shown in Fig.6(a) as well. In addition, and relating to your attached figure where you attempted to reproduce the figure, for each combination we also get a different number density after having assumed a certain sigma and retrieved a corresponding median radius value. This means, each curve in Fig.6(a) also has a different number density. In other words: You reproduced the blue and red curves from the figure correctly, albeit with a fixed number density. To avoid misunderstandings by the reader we added another sub-figure to Fig.6. Here, we show the same PSDs as in Fig. 6(a), but scaled to the same number density value to facilitate easier comparison of the individual curves. We explain this in the text and in the caption.

**Comment:** 296 Just to be clear suggest … Each curve corresponds to a single σ value …

**Answer/Changes:** We implemented it.

**Comment:** 310-312 This observation about the distribution width is a bit misleading, since it only appears in distributions of dn/dr. It does not appear in distributions of dn/dln(r), then the widths do appear as expected with σ near 1.0 being narrow and near 2.0 much broader, and each distribution is centered on the median radius, instead of drifting with σ due to the inherent 1/r in the dn/dr distribution. Recall these are lognormal distributions not normal distributions.

**Answer/Changes:** We now deleted this passage, since we instead included a new figure following your suggestion of one of your comments from above where we comment on this. In this figure we illustrate the relationship between sigma and omega and also communicate clearer the distinction between linear and log radius space.

**Comment:** 370-375 This explanation of how the comparison was conducted between model and measurements is the best way to do it; however, Fig. 7 is not entirely consistent with this explanation, particularly Fig. 7d). Note the differences between model estimates below the tropopause, particularly in the northern hemisphere, compared to the measurements. Including these in the model results distracts the reader from the comparison's salient points, and the authors make no mention of this region in their discussion.

**Answer/Changes:** Thank you for spotting this difference, this was an oversight on our side. We now updated Fig. 7 to treat values below the tropopause the same in both the satellite retrieval and the model simulation data.

**Comment:** 390-395 The inclusion of m3/m2 is rather superfluous. The important quantities are V and A, however they are calculated. Here the authors also need to be clear that there is a difference of a factor of three between the measurements (which do not include the factor of 3) and the model (which do) when comparing the effective radii using Eqns (2) and (4).

**Answer/Changes:** See our answer to your comment on line 102. There is no factor 3 difference between the effective radii calculated from the SAGE III/ISS retrieval data and the effective radii calculated from the ECHAM model simulations. We acknowledge that there is some potential for confusion, which is why we removed the "V/A *3" from equation (4) instead of deriving it from the equations given in Grainger (2022). The model puts out the third and second moments of the size distributions and from that the effective radii are calculated.

**Comment:** 396-424 Based on the comparison in Fig. 8, it appears that the discrepancy of the factor of three has been accounted for by the authors. At least one would hope so. But until this distinction is made clear it is difficult to fully trust this comparison.

**Answer/Changes:** See the answer to the comment above. There is no factor 3 discrepancy. We suspect, that there may also have been a misunderstanding, because the rightmost plots of Fig.8, i.e. Fig.8 (c) and (f), do not depict the same quantity as the other plots, but instead show the difference between the effective radii from August and June 2019. To make this clearer, we adjusted the title text of these plots.

**Referee #2 - Daniele Visioni comments:**

**Comment:** This is a very nice and interesting new work looking at recent smaller volcanic eruption with a mix of observations and modeling with ECHAM and I endorse its publication. It is nicely written, clear (except in a very few parts!), and the analyses are rather in depth and well discussed.

I agree with all of the comments from reviewer 1, so the authors have to address those (especially the discrepancy of Fig. 6 compared to what they showed).

**Answer:** Thank you very much for the kind words as well as the constructive criticism. We gladly picked up your comments, implemented your suggestions and addressed the comments by referee 1.

**Comment:** A few more comments from me:

L 91: are you sure (Oxford, 2018) is the correct reference?? Oxford is the name of the university, not the authors?

**Answer/Changes:** We did not find a list of authors to cite and other published papers citing these Mie routines also don't cite authors. We still changed our citation to what was similarly used in other papers before:

Mie scattering routines: Oxford University - Department of Physics - Earth Observation Data Group: Mie Scattering Routines, Accessed August 20, 2018 at: http://eodg.atm.ox.ac.uk/MIE/index.html .

**Comment:** L 90-95 needs to be rewritten for clarity. "a kind of two-dimensional lookup-table"?

**Answer/Changes:** We rewrote this section for clarity in response to this comment as well as to a comment by referee 1.

**Comment:** L 117: "Above the tropopause, a simple stratospheric sulfur chemistry was applied" replace "was applied" with "was considered".

**Answer/Changes:** We implemented the suggestion.

**Comment:** L 118: "oxidant fields" would only refer to OH, no? So better just say "chemical species".

**Answer/Changes:** Implemented.

**Comment:** L 244: "overwritten" is not the proper word here. "Covered" or "dwarfed" is better.

**Answer/Changes:** We changed it to "covered".

**Comment:** L 255: sometimes the authors could use slightly more scientific terms… "Therefore it is not out of the question whether smoke may have played a role" is very qualitative, doesn't add anything, and is a very long-winded way to say "It cannot be excluded that smoke might have played a role".

**Answer/Changes:** Thank you for spotting this. We changed it according to your suggestion.

**Comment:** L 281: "varies in time and space".

**Answer/Changes:** Implemented

**Comment:** Ll 290-292: this phrase is rather obscure and should be rewritten.

**Answer/Changes:** We split the phrase into two sentences and rephrased it to make it clearer:

"All depicted PSDs, including the red and blue curves, represent possible solutions if the aerosol size retrieval was performed using only two wavelengths. These solutions are very different and we would not know which one is closest to the truth, therefore we could not conclude in which way the size distribution of the stratospheric aerosol changes after volcanic eruptions, when using only two wavelengths."

**Comment:** L 258: I would use "considered" rather than "implemented".

**Answer/Changes:** We assume you are referencing line 358 and changed it to "considered".

**Comment:** L 416: Again, "destroy the signal" is not very scientific.

**Answer/Changes:** We changed it to "cover the signal".

**Comment:** L 420: "in principle"

**Answer/Changes:** Thanks for spotting the typo. We corrected it.

**Comment:** Ll 415-423: One could suggest a few (future of course) tests for these assumed sources of discrepancy: for instance, using runs with prescribed meteorology versus free-running could highlight if the problem is a "deviation in dynamics" as opposed to microphysical uncertainties. Also, the lack of interaction between ash and sulfate, which is mentioned elsewhere, and of interactive chemistry, could also be sources that should be mentioned here. Apparently they do not affect the initial plume that much, but they might later on.

**Answer/Changes:** Thank you for the suggestions. We don't think that ash may play a role here as ash is important in the early phase of the plume evolution. We assume that differences between ECHAM-HAM and the observations are in the microphysics and/or the lifetime of OH. A comparison with other models would be very helpful here. We changed the segment and do now mention this need for more comparison.

**Comment:** L 456: which projections? You can be a bit more specific here. For instance, one could derive some suggestions on the average stratospheric size distribution used for future CMIP projections, considering these volcanism episodes are frequent.

**Answer/Changes:** Our wording was unclear here. We changed it from "projections" to "simulations" to avoid misunderstandings.

Sources mentioned:

Wrana, F., von Savigny, C., Zalach, J., Thomason, L. W.: Retrieval of stratospheric aerosol size distribution parameters using satellite solar occultation measurements at three wavelengths, Atmos. Meas. Tech., 14, 2345 - 2357, doi: 10.5194/amt-14-2345-2021, 2021.

Grainger, R. G.: Some Useful Formulae for Aerosol Size Distributions and Optical Properties, 2017, accessed August 8th, 2022 at: http://eodg.atm.ox.ac.uk/user/grainger/research/aerosols.pdf .